# Cross talk between glucose metabolism and immunosuppression in IFN-γ–primed mesenchymal stem cells

Mengwei Yao[1,2,3,*] , Zhuo Chen[1,2,*], Xiao He[4] , Jiaoyue Long[1,2], Xuewei Xia[1,2,3], Zhan Li[1,2], Yu Yang[1,2], Luoquan Ao[1,2], Wei Xing[1,2], Qizhou Lian[5,6,7], Huaping Liang[1] , Xiang Xu[1,2,3]

**The immunosuppressive function "licensed" by IFN-γ is a vital attribute of mesenchymal stem cells (MSCs) widely used in the treatment of inflammatory diseases. However, the mechanism and impact of metabolic reprogramming on MSC immunomodulatory plasticity remain unclear. Here, we explored the mechanism by which glucose metabolism affects the immunomodulatory reprogramming of MSCs "licensed" by IFN-γ. Our data showed that glucose metabolism regulates the immunosuppressive function of human umbilical cord MSCs (hUC-MSCs) challenged by IFN-γ through the Janus kinase–signal transducer and activator of transcription (JAK-STAT) pathway. Furthermore, ATP facilitated the cross talk between glucose metabolism and the JAK-STAT system, which stimulates the phosphorylation of JAK2 and STATs, as well as the expression of indoleamine 2, 3-dioxygenase and programmed cell death-1 ligand. Moreover, ATP synergistically enhanced the therapeutic efficacy of IFN-γ–primed hUC-MSCs against acute pneumonia in mice. These results indicate a novel cross talk between the immunosuppressive function, glucose metabolism, and mitochondrial oxidation and provide a novel targeting strategy to enhance the therapeutic efficacies of hUC-MSCs.**

## Introduction

Mesenchymal stem cells (MSCs) are multipotent adult stem cells that are widely distributed in multiple tissues, including the umbilical cord, placenta, bone marrow, and fat (Brown et al, 2019). In addition, human mesenchymal stem cells (hMSCs) are considered an off-the-shelf, universal cellular product for cell therapy. They are extensively used in treating various diseases, owing to their tissue repair and regenerative properties in addition to their strong immunosuppressive and anti-inflammatory effects (Ma et al, 2014; Luo et al, 2019). At present, more than 1,200 clinical trials on hMSCs have been registered on Clinicaltrials.gov to evaluate the safety and efficacy of hMSCs for the treatment of various diseases (Wobma et al, 2018a; 2018b). Besides their properties of tissue repair and regeneration, MSCs also display strong systemic immunosuppressive effects through multiple mechanisms. Thus, MSC-based cell therapy is considered a promising strategy for treating autoimmune or inflammatory diseases, such as graft-versus-host disease, inflammatory bowel disease, multiple sclerosis, and coronavirus disease 2019 (COVID-19) (Ma et al, 2014; Li et al, 2019; Luo et al, 2019; He et al, 2020; Zhu et al, 2021).

However, the current clinical efficacy of hMSCs is still variable, even among individuals with the same autoimmune disease (Sudres et al, 2006; Galipeau & Sensébé, 2018; Yang et al, 2018). Therefore, it is imperative to deepen our understanding of the immunomodulatory mechanism of hMSCs and explore new strategies to improve their immunosuppressive and anti-inflammatory functions. Many studies have shown that the immunosuppressive capacity of MSCs not only depends on soluble factors such as indoleamine 2, 3-dioxygenase (IDO) secreted by MSCs but also involves expressing immunosuppressive receptors such as programmed cell death-1 ligand (PD-L1) (Meisel et al, 2004; Ren et al, 2009; Li et al, 2019). This allows direct interaction with various types of innate and adaptive immune cells and inhibits their proliferation and activation (Sotiropoulou et al, 2006; Spaggiari & Moretta, 2013). Nevertheless, the immunosuppressive role of MSCs is not constitutive but needs to be "licensed" by the cytokine-mediated immunological microenvironment, and this depends on the reprogramming of the plasticity and function of MSCs (Li et al, 2012; Krampera et al, 2013).

---

[1]Department of Stem Cell and Regenerative Medicine, Daping Hospital, Army Medical University, Chongqing, China    [2]Central Laboratory, State Key Laboratory of Trauma, Burn and Combined Injury, Daping Hospital, Army Medical University, Chongqing, China    [3]Department of Biochemistry and Molecular Biology, College of Basic Medical Sciences, Army Medical University, Chongqing, China    [4]PLA Rocket Force Characteristic Medical Center, Beijing, China    [5]HKUMed Laboratory of Cellular Therapeutics, and State Key Laboratory of Pharmaceutical Biotechnology, The University of Hong Kong, Pok Fu Lam, Hong Kong    [6]Cord Blood Bank, Guangzhou Women and Children's Medical Center, Guangzhou Medical University, Guangzhou, China    [7]Department of Surgery, The University of Hong Kong Shenzhen Hospital, Shenzhen, China

Correspondence: xiangxu@tmmu.edu.cn; 13638356728@163.com
*Mengwei Yao and Zhuo Chen contributed equally to this work.

One of the cytokines, IFN-γ, is regarded as a key inducer of MSC immunosuppressive/anti-inflammatory functions (Liu et al, 2011; Yang et al, 2018). IFN-γ licenses immunosuppressive MSCs by activating the Janus kinase–signal transducer and activator of transcription (JAK-STAT) pathway to up-regulate the expression of IDO and PD-L1 (Hammarén et al, 2015). We previously demonstrated that a combination of MSCs and IFN-γ in vivo markedly improves the clinical efficacy of MSC-based therapy in rheumatoid arthritis patients (He et al, 2020). However, it was reported that IFN-γ–licensed hMSCs did not exhibit anti-inflammatory effects or clinical efficacy as expected in Crohn's disease patients (Taddio et al, 2015). Sometimes, hMSCs display an immune-promoting effect when pro-inflammatory cytokines (including IFN-γ, TNF-α, and IL-1α/β) are inadequate to elicit sufficient nitric oxide (Li et al, 2012). The above evidence suggests that reprogramming MSC immunosuppressive function is a complex process regulated by various factors and pathways and needs further exploration.

Glucose metabolism is the basis for sustaining cell functions and can provide energy and substrates for cell activities. Many studies have shown that glucose metabolic reprogramming regulates polarization in macrophages, microglia, and T cells (Almeida et al, 2016; O'Neill & Pearce, 2016). In addition, recent advances have shown that in MSCs, glucose metabolic characteristics are related to cell proliferation, lineage-specific differentiation, and their in vivo origin (Liu & Ma, 2015). However, whether and how glucose metabolic reprogramming regulates MSC immunosuppression in response to IFN-γ stimulation remains unclear.

Moreover, IFN-γ enhances glycolysis in human bone marrow–derived MSCs (hBM-MSCs). Inhibition of mitochondrial oxidative phosphorylation promotes the anti-inflammatory function of hBM-MSCs (Liu et al, 2019). It is unknown whether human umbilical cord mesenchymal stem cells (hUC-MSCs) with more extensive clinical sources and low immunogenicity also have the same glucose metabolism pattern. Current evidence shows that MSCs from different tissues have significant differences in certain functions (such as proliferation, differentiation, and immunoregulation); however, it is unclear whether these differences are also reflected in the glucose metabolism pattern. To address the relationship and regulatory mechanism between the glucose metabolism and immunosuppressive function of hUC-MSCs, we explored the glucose metabolism pattern in hUC-MSCs "licensed" by IFN-γ and found that its metabolic pattern was significantly different from that of hBM-MSCs, displaying weak glycolysis and increased aerobic oxidation. The expression of immunosuppressive factors, such as IDO and PD-L1, was significantly reduced upon inhibiting the activity of key enzymes of aerobic oxidation in hUC-MSCs but not affected by the inhibition of lactate dehydrogenase (LDH), a key enzyme of lactate glycolysis. Mechanistically, the aerobic oxidative metabolism of glucose modulates the immunosuppressive function of IFN-γ–licensed hUC-MSCs (IFN-γ–hUC-MSCs) by increasing STAT phosphorylation. This phosphorylation is also synergistically enhanced in vitro and in vivo by ATP.

Overall, we found that glucose metabolic reprogramming is a novel modulatory mechanism for the immunosuppressive function of IFN-γ–challenged hUC-MSCs. Thus, our study provides evidence that ATP facilitates the "licensing" effect of IFN-γ in hUC-MSCs and broadens the current molecular understanding of the immunomodulatory plasticity of MSCs.

# Results

## IFN-γ licenses the immunosuppressive reprogramming of hUC-MSCs

To confirm that IFN-γ licenses the immunosuppressive reprogramming of hUC-MSCs, we investigated the anti- and pro-inflammatory gene expression profile in IFN-γ–primed hUC-MSCs and their immunosuppressive capacity on T-cell proliferation in vitro. The results showed that IFN-γ treatment significantly up-regulated the expression of anti-inflammatory factors such as IDO, CD274 (PD-L1), secretory leucoprotease inhibitor (SLPI), and interleukin-18–binding protein (IL18BP) (Fig 1A). Interestingly, IFN-γ had different effects on the expression of pro-inflammatory factors. It promoted the expression of TNF, IL-6, IL-7, and C-X-C motif chemokine ligand 8. In contrast, it inhibited the expression of interleukin 1 receptor type 1 (IL1R1), transforming growth factor-beta 1 (TGFB1), and transforming growth factor-beta 3 (TGFB3) (Fig 1B).

Nevertheless, to verify the total impact of IFN-γ on the immunomodulatory function of hUC-MSCs at the functional level, we tested the effect of IFN-γ–treated hUC-MSCs on T-cell proliferation. The results indicated that IFN-γ significantly inhibited the proliferation of T cells in hUC-MSCs (Fig 1C and D). In addition, the mean fluorescence index and cell counts of T cells after co-cultured with hUC-MSCs indicated the same (Fig S1A and B). These data suggest that IFN-γ induces the immunosuppressive reprogramming of hUC-MSCs.

## IFN-γ treatment reconfigures the energy metabolism of hUC-MSCs toward aerobic oxidation

To explore whether IFN-γ treatment resulted in the metabolic reconfiguration of hUC-MSCs, we analyzed the content of glucose metabolism intermediates, expression of key glucose metabolism-related enzymes, extracellular acidification rate (ECAR), and oxygen consumption rate (OCR) in IFN-γ–primed hUC-MSCs. The results showed that IFN-γ treatment influenced the glucose metabolism of hUC-MSCs. The levels of important intermediates were significantly reduced (Fig 2A and B). We further found that the expression of key glucose metabolism–related enzymes changed significantly. Hexokinase 2 (HK2) and enolase 1 (ENO1) were up-regulated. In contrast, pyruvate dehydrogenase E1 subunit alpha 1 (PDHA1), pyruvate dehydrogenase E1 subunit beta (PDHB), pyruvate dehydrogenase kinase 1 (PDK1), and lactate dehydrogenase D (LDHD) were down-regulated (Fig 2C and D). Importantly, the down-regulated enzymes, including PDHA1, PDHB, and PDK1, are closely related to the regulation of pyruvate dehydrogenase (PDH). Therefore, PDH may have a critical influence on the regulation of glucose metabolism in IFN-γ–primed hUC-MSCs.

To determine whether IFN-γ affected aerobic or anaerobic glucose metabolism, we evaluated the metabolic status of IFN-γ–treated and untreated hUC-MSCs in real-time. We found that IFN-γ treatment promoted aerobic oxidation (Fig 2E, F, and I) but inhibited anaerobic glycolysis (Fig 2G and H). Moreover, we demonstrated that the stimulated culture media of OCR and ECAR had no influence on glycolytic capacity (Fig S2A and B). These results suggest

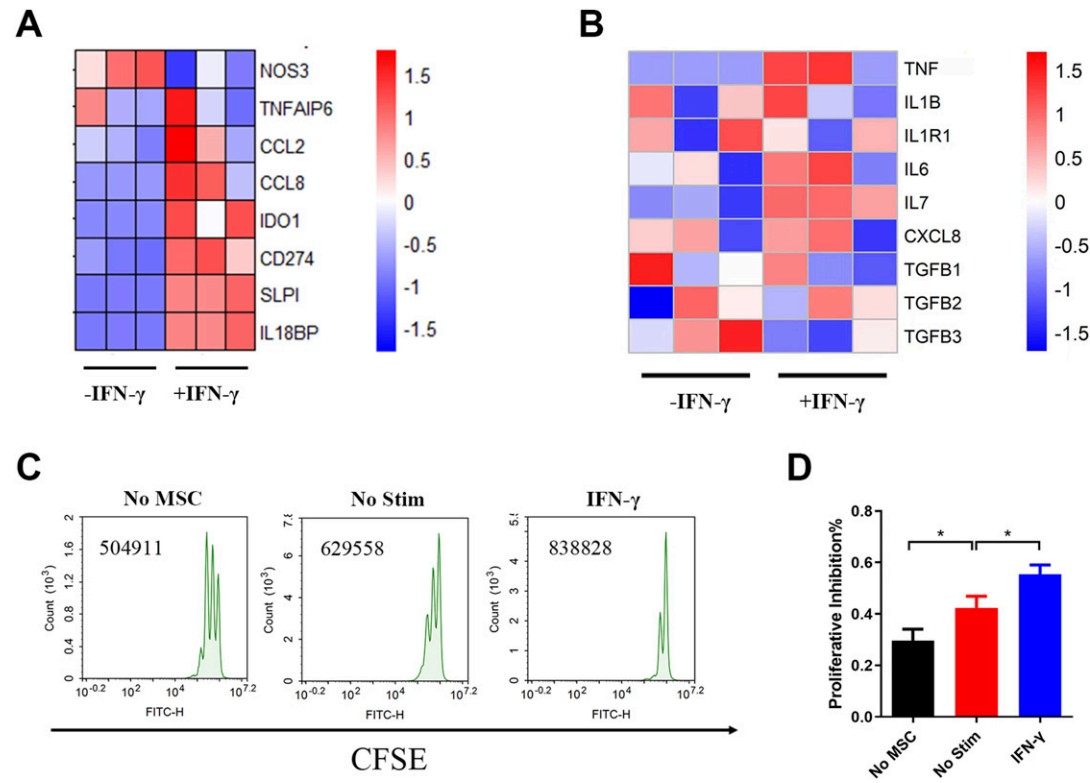

**Figure 1. IFN-γ treatment licenses the immunosuppressive reprogramming of hUC–mesenchymal stem cells (MSCs).**
**(A, B)** Heat map of transcriptome sequencing for anti-inflammatory (A) and pro-inflammatory (B) cytokines. **(C, D)** Proliferation assay of T cells co-cultured with hUC-MSCs in vitro. Data are represented as mean ± SD (n = 3, per group). *P < 0.05, using one-way ANOVA. No MSC, single PBMC group; No Stim, untreated hUC-MSCs co-cultured with the PBMC group; IFN-γ, hUC-MSCs pretreated by IFN-γ co-cultured with the PBMC group.
Source data are available for this figure.

that IFN-γ treatment reconfigures the energy metabolism of hUC-MSCs toward aerobic oxidation and affects the metabolic rate.

### Aerobic oxidation is required to express IDO and PD-L1 in IFN-γ–primed hUC-MSCs

Previous studies suggest that glucose metabolism regulates the anti-inflammatory function of MSCs (Jitschin et al, 2019; Liu et al, 2019; Vigo et al, 2019). Glucose metabolism involves lactic acid glycolysis and aerobic oxidation. In addition, MSCs from different tissues, organs, and species are functionally distinguished (Wobma et al, 2018a; 2018b). Therefore, to determine which pathways affect the anti-inflammatory function of hUC-MSCs, we explored the effects of key enzymes in glucose metabolism on the secretion of critical anti-inflammatory factors IDO and PD-L1 in hUC-MSCs primed by IFN-γ.

The results indicated that the use of 2-deoxy-D-glucose (2-DG) to inhibit the entire glucose metabolism or the use of siRNA to inhibit PDH (Fig S3A–C) significantly reduced the protein levels of IDO and PD-L1, whereas inhibiting the activity of LDH did not affect their protein levels (Fig 3A–C). In addition, when rotenone was used to inhibit mitochondrial function, it inhibited the expression of IDO and PD-L1 in hUC-MSCs induced by IFN-γ (Fig 3D–F). This provides evidence that cytosolic glycolysis is not relevant, whereas mitochondrial respiration either from glucose or fatty acid oxidation is essential for the anti-inflammatory function of IFN-γ–primed hUC-

MSCs. The inhibition of aerobic oxidation also resulted in the inhibition of IDO and PD-L1 production.

### IFN-γ–induced metabolic reconfiguration activates JAK-STAT signaling in hUC-MSCs

IFN-γ promotes nuclear translocation by inducing the early phosphorylation of STAT1 (Vigo et al, 2017), which up-regulates the expression of anti-inflammatory mediators such as PD-L1 and IDO in MSCs (Mounayar et al, 2015; Mimura et al, 2018). In addition, glucose metabolism influences the JAK-STAT1 pathway during macrophage polarization. Inhibition of glycolysis attenuates IFN-γ–induced STAT1 phosphorylation, thereby inhibiting the polarization of M1 macrophages (Wang et al, 2018).

To confirm the role of glucose metabolism in the phosphorylation of JAK-STAT in hUC-MSCs, we investigated the effects of key enzymes in glucose metabolism on the phosphorylation of STAT1–3. The results indicated that aerobic oxidation was positively correlated with the phosphorylation of STAT1–3. Inhibiting the activity of key enzymes in aerobic oxidation decreased the phosphorylation of STAT1–3, whereas inhibiting the activity of LDH had little effect on the phosphorylation of STAT1–3. Interestingly, phosphorylation of STAT1–3 had different sensitivities with respect to changes in glucose metabolism. LDH-IN-1 inhibited the phosphorylation of STAT2 but did not affect the phosphorylation of STAT1 or STAT3

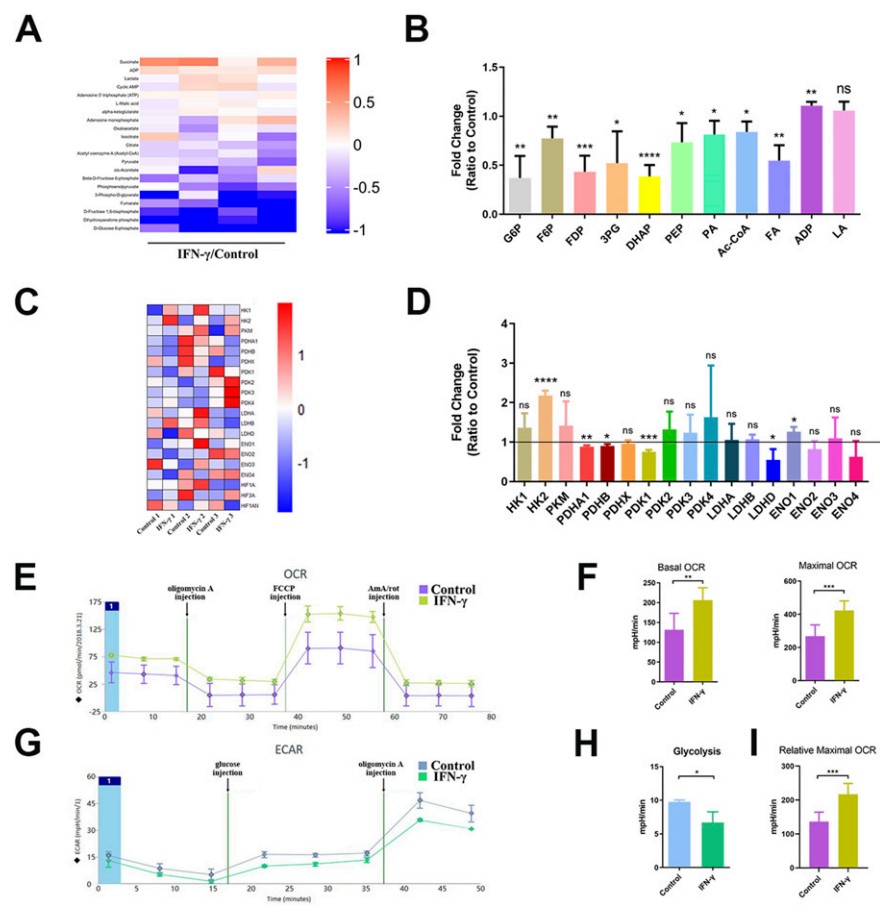

**Figure 2. IFN-γ treatment reconfigures the energy metabolism of hUC–mesenchymal stem cells (MSCs) toward aerobic oxidation.**
**(A, B)** LC–MS detected the changes in the intermediates of glucose metabolism in hUC-MSCs untreated or treated with IFN-γ (40 ng/ml) for 24 h (n = 4, mean ± SD). ns: not significant, *P < 0.05, **P < 0.01, ***P < 0.001, ****P < 0.0001, using t test. **(C, D)** Transcriptome sequencing analyzed the expression of key enzymes of glucose metabolism in hUC-MSCs untreated or treated with IFN-γ (40 ng/ml) for 24 h (n = 3, mean ± SD). ns, not significant, *P < 0.05, **P < 0.01, ***P < 0.001, ****P < 0.0001, using t test. **(E, F, G, H, I)** ECAR and oxygen consumption rate profiles of hUC-MSCs were measured in the presence and absence of IFN-γ (n = 3, mean ± SD). ns, not significant, *P < 0.05, **P < 0.01, ***P < 0.001, using t test.
Source data are available for this figure.

(Fig 4A–D). This indicates that the metabolic regulatory mechanism of STAT1–3 phosphorylation is distinct. Phospho-STAT2 was affected by entire glucose metabolism, whereas phospho-STAT1 and phospho-STAT3 were only affected by aerobic oxidation.

### ATP enhances the immunosuppressive capacities of IFN-γ–primed hUC-MSCs by activating the JAK-STAT pathway

Previous studies confirmed that JAK is an ATP-dependent protein kinase. ATP can bind to JAK2 and specifically regulate the phosphorylation of STAT, ultimately affecting cellular immune regulation, metabolism, and apoptosis mediated by the JAK-STAT pathway. When the ATP-binding site of JAK2 is mutated, it causes excessive activation of JAK2 and changes cell-related biological functions (Hammarén et al, 2015). Therefore, we investigated the role of ATP in the expression of IDO and PD-L1 in IFN-γ–primed hUC-MSCs.

As shown in the results, although IFN-γ treatment reduced the ATP levels, the difference was not significant, indicating a homeostatic mechanism that promptly supplemented the ATP consumption caused by protein synthesis induced via IFN-γ. When aerobic oxidation was inhibited by PDH siRNA, IFN-γ treatment markedly reduced the ATP levels, which proved the vital role of ATP in the "licensing" effect of IFN-γ in hUC-MSCs (Fig 5J). Otherwise, IFN-γ and ATP have no influence on cell survival (Fig S4A–C). Next, we further verified the effect of ATP on the phosphorylation of JAK2 and STAT1-3 and the expression of

PD-L1 and IDO. It was found that adding a certain concentration of ATP (2 mM) to hUC-MSCs before IFN-γ pretreatment dramatically promoted the expression of IDO and PD-L1 and the phosphorylation of JAK2 and STAT1-3 (Fig 5A–G). Notably, the promotion effect of ATP was not instantaneous (0.5 and 1 h) but long-term (24 h). ATP even promoted the phosphorylation of JAK2 without IFN-γ induction.

To further assess the effects of hUC-MSCs pretreated with a combination of ATP and IFN-γ on the inhibition of T-cell proliferation, we co-cultured hUC-MSCs with human peripheral blood mononuclear cells (hPBMCs) (Fig 5H and I). Compared with the untreated hUC-MSCs, IFN-γ–treated hUC-MSCs inhibited T-cell proliferation, and ATP synergistically enhanced the promotion effect of IFN-γ; thus, hUC-MSCs had a better T-cell proliferation–inhibitory function. The mean fluorescence index and cell counts of T cells after co-cultured with hUC-MSCs also corroborated the above results (Fig S5A and B). These data indicate that ATP promoted the phosphorylation of STAT1–3 to enhance the expression of IDO and PD-L1 in IFN-γ–primed hUC-MSCs, a phenomenon that eventually promoted its inhibitory effects on T-cell proliferation.

### ATP synergistically enhanced the therapeutic effect of IFN-γ–primed hUC-MSCs transplantation for acute pneumonia in mice

To further test the function of hUC-MSCs synergistically primed by ATP and IFN-γ in vivo, we used a mouse model of acute pneumonia to evaluate

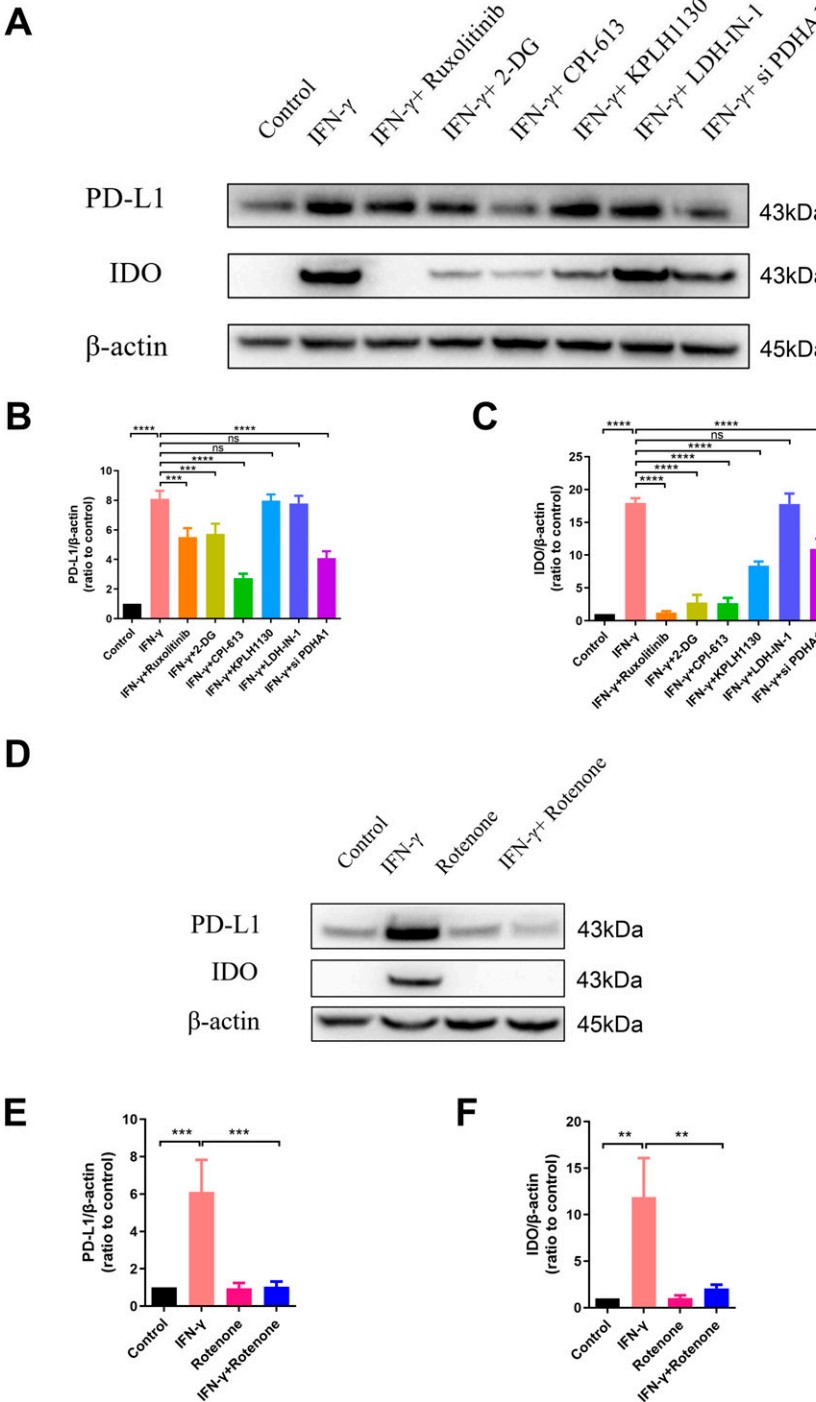

**Figure 3. Aerobic oxidation is required to express IDO and PD-L1 in IFN-γ–primed hUC–mesenchymal stem cells.**
**(A, D)** Immunoblot analysis of specified proteins in hUC–mesenchymal stem cells after stimulation with IFN-γ ± 1 h pretreatment with ruxolitinib, 2-DG, CPI-613, KPLH1130, LDH-IN-1, PDHA1 siRNA, and rotenone. Ruxolitinib, an inhibitor of the JAK-STAT pathway; 2-DG, an inhibitor of hexokinase; CPI-613, an inhibitor of pyruvate dehydrogenase and α-ketoglutarate dehydrogenase; KPLH1130, an inhibitor of pyruvate dehydrogenase kinase; LDH-IN-1, an inhibitor of lactate dehydrogenase; rotenone, an inhibitor of mitochondrial electron transport chain complex I. β-actin served as a loading control. **(B, C, E, F)** Relative densitometric quantitation of the shown blot, after normalization on actin (n = 3, mean ± SD). ns, not significant, **$P < 0.01$, ***$P < 0.001$, ****$P < 0.0001$, using one-way ANOVA.
Source data are available for this figure.

their therapeutic efficacy. After dropping LPS on the posterior pharyngeal wall for 4 h, the mice received a caudal vein injection of $1 × 10^6$ hUC-MSCs or PBS. Subsequently, the mice were euthanized on day 3 (Fig 6A).

The results of hematoxylin and eosin (HE) staining of the lung tissue are shown in Fig 6B and C. The results exhibited almost no complete alveolar structure in the acute pneumonia mice. Concurrently, hUC-MSC treatment maintained the alveolar structure, and IFN-γ combined with ATP treatment had better effects compared with IFN-γ treatment alone.

Inflammatory cell infiltration of the alveolar lavage fluid is shown in Fig 6D. Normal mice had almost no inflammatory cell infiltration, whereas acute pneumonia mice had increased macrophage, neutrophil, and T-cell infiltration. After treatment with hUC-MSCs, the alveolar inflammatory cell infiltration was significantly reduced. Notably, hUC-MSCs combined with ATP and IFN-γ had a better curative effect, almost reaching the level observed in normal mice.

IL-1β and TNF-α are strongly induced by LPS, the content of which is also an indicator of inflammation (Guglielmotti et al, 1997).

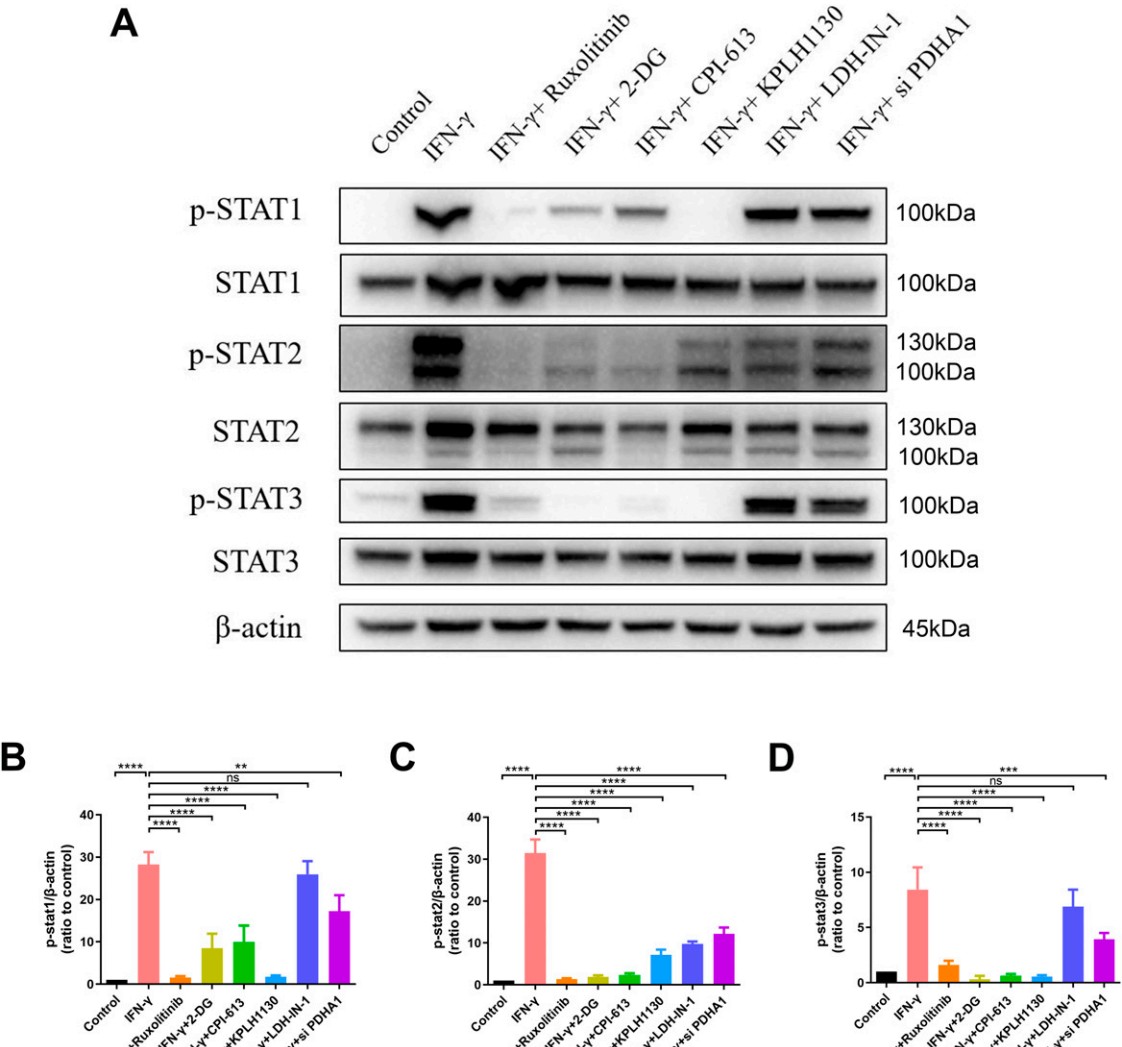

**Figure 4. IFN-γ–induced metabolic reconfiguration activates JAK-STAT signaling in hUC–mesenchymal stem cells.**
**(A)** Immunoblot analysis of specified proteins in hUC–mesenchymal stem cells after stimulation with IFN-γ ± 1 h pretreatment with ruxolitinib, 2-DG, CPI-613, KPLH1130, LDH-IN-1, PDHA1 siRNA, and rotenone. β-actin served as a loading control. **(B, C, D)** Relative densitometric quantitation of the shown blot, after normalization on actin (n = 3, mean ± SD). ns: not significant, **$P < 0.01$, ***$P < 0.001$, ****$P < 0.0001$, using one-way ANOVA.
Source data are available for this figure.

Therefore, we measured the concentrations of IL-1β and TNF-α in lung tissue and alveolar lavage fluid. As shown in Fig 6E–H, hUC-MSC treatment reduced IL-1β and TNF-α levels in acute pneumonia mice. In comparison to IFN-γ treatment alone, the combined treatment of ATP and IFN-γ lowered the levels of IL-1β and TNF-α. This further confirmed that ATP combined with IFN-γ pretreatment had better therapeutic efficacy than IFN-γ treatment alone.

## Discussion

MSCs display strong immunosuppressive and anti-inflammatory effects and can induce stable host immune tolerance through multiple mechanisms. Thus, MSC-based cell therapy is currently considered a promising treatment strategy for autoimmune and inflammatory diseases (Wang et al, 2014). However, such immunomodulatory functions of MSCs are not innate and immutable but dependent on the induction of the inflammatory microenvironment (Li et al, 2012; Krampera et al, 2013). There are substantial variations in the immune microenvironment among different stages of the disease or in different patients, which may affect the therapeutic efficacy of MSCs. The immunosuppressive functions of MSCs are reported to be licensed by IFN-γ, together with the simultaneous presence of three pro-inflammatory cytokines, TNF-α, IL-1α, and IL-1β. The expression of several chemokines, IDO and PD-L1, is upregulated by activating the JAK-STAT pathway (Mounayar et al, 2015; Mimura et al, 2018). Consistent with the above results, our previous clinical studies found that patients with high IFN-γ levels or those in receipt of IFN-γ treatment responded well to MSCs, whereas

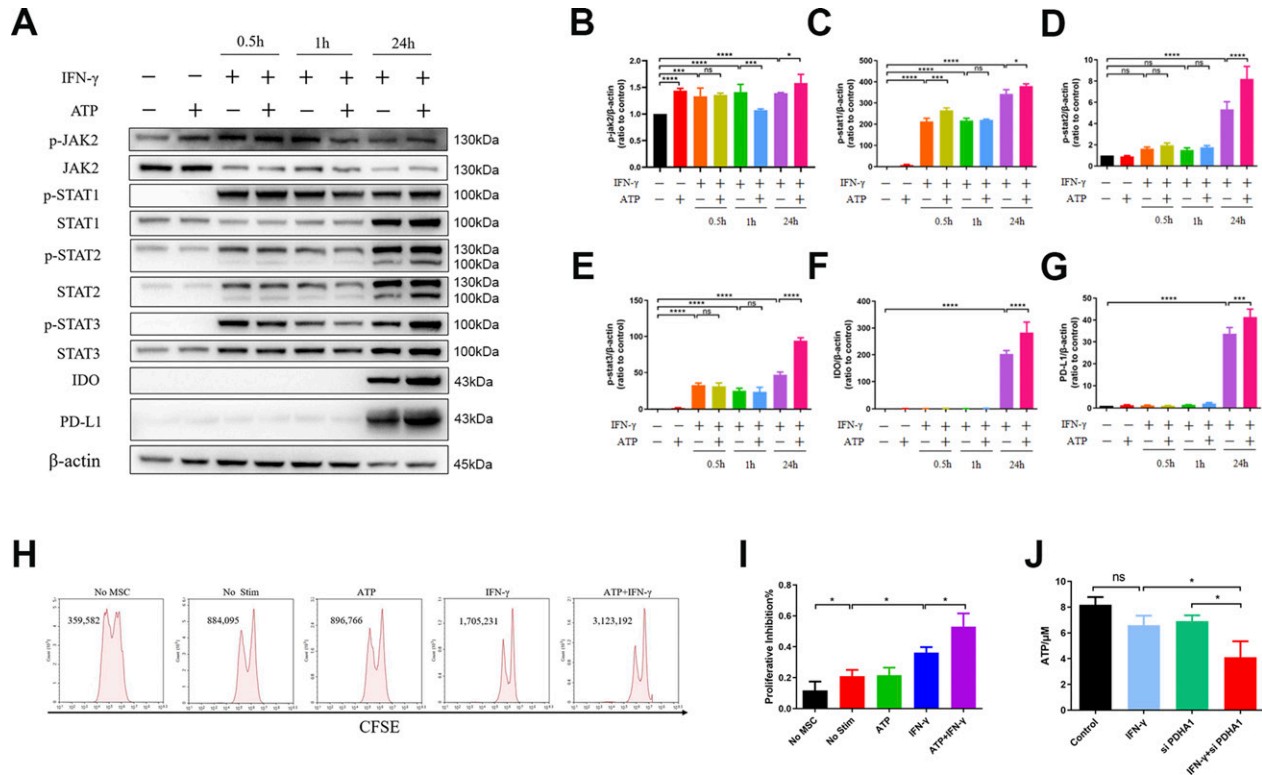

**Figure 5. ATP enhances the immunosuppressive capacities of IFN-γ–primed hUC–mesenchymal stem cells (MSCs) by activating the JAK-STAT pathway.**
**(A)** Immunoblot analysis of specified proteins in hUC-MSCs. **(B, C, D, E, F, G)** Relative densitometric quantitation of the shown blot, after normalization on actin (n = 3, mean ± SD). ns, not significant, *P < 0.05, ***P < 0.001, ****P < 0.0001, using one-way ANOVA. **(H, I)** Proliferative inhibition assay of T cells co-cultured with hUC-MSCs in vitro. Data are represented as mean ± SD, *P < 0.05, using one-way ANOVA. No MSC, single PBMC group; No Stim, untreated hUC-MSCs co-cultured with the PBMC group; ATP, ATP-incubated hUC-MSCs co-cultured with the PBMC group; IFN-γ, hUC-MSCs pretreated with IFN-γ co-cultured with the PBMC group; ATP + IFN-γ, hUC-MSCs pretreated with IFN-γ and ATP co-cultured with the PBMC group. **(J)** ATP production of hUC-MSCs after stimulation with IFN-γ ± 15 min pretreatment with PDHA1 siRNA (n = 3, mean ± SD). ns: not significant, *P < 0.05, using one-way ANOVA.
Source data are available for this figure.

patients with low IFN-γ levels responded poorly (Yang et al, 2018; He et al, 2020). In this study, IFN-γ alone treatment in vitro also licenses the immunosuppressive reprogramming of hUC-MSCs (Figs 1 and S1).

The central energy metabolism exerts a significant role in cellular immunomodulation (Almeida et al, 2016; O'Neill & Pearce, 2016; Wang et al, 2018). It has been demonstrated that IFN-γ can induce a metabolic switch to aerobic glycolysis in hBM-MSCs, whereas inhibiting mitochondrial metabolism up-regulates the expression of immunosuppressive factors IDO and prostaglandin E₂. This enhances the immunosuppressive or anti-inflammatory function of hBM-MSCs (Liu et al, 2019). However, in our study, IFN-γ–licensed hUC-MSCs exhibited different metabolic patterns, increased aerobic oxidation, and decreased glycolysis, which reconfigured glucose metabolism to an aerobic oxidative phenotype (Figs 2 and S2). In addition, aerobic oxidation is required to express IDO and PD-L1 in IFN-γ–primed hUC-MSCs (Figs 3 and S3). Moreover, MSCs from various tissues possess significant functional heterogeneity, such as the immune phenotype, cytokine secretion profile, and immunomodulatory activity (Wobma et al, 2018a; 2018b). Therefore, our data provide further evidence that MSCs from different tissues have heterogeneity in glucose metabolism, which

should be considered when using MSCs in clinical practice. Further exploration is required to elucidate the other metabolic changes in energy metabolism.

Although glucose metabolism has been shown to affect the immunosuppressive function of MSCs, to the best of our knowledge, no studies have clarified the underlying molecular mechanisms. Our investigation suggests that the metabolic intermediate ATP can synergistically promote the "licensing" effect of IFN-γ on hUC-MSCs to enhance phosphorylation of the JAK-STAT pathway (Figs 4 and 5). The immunosuppressive effects of IFN-γ treatment were synergistically enhanced in vitro and in vivo by ATP in hUC-MSCs (Figs 5, 6, S4, and S5). It is still necessary to investigate how ATP further regulates STAT phosphorylation. A previous study has shown that ATP can bind to a pseudokinase domain in JAK2 and participate in the hyperactivation of JAK2. Mutation of the amino acid at the binding site ameliorates hyperphosphorylation of JAK2 in mutants V617F (Hammarén et al, 2015). Therefore, we speculated that the phosphate group might be transferred to STAT from ATP when it binds to JAK2.

Finally, we proved that pretreatment with ATP combined with IFN-γ was safe and effective in a mouse model of acute pneumonia as evidenced by the significant reduction in alveolar inflammatory

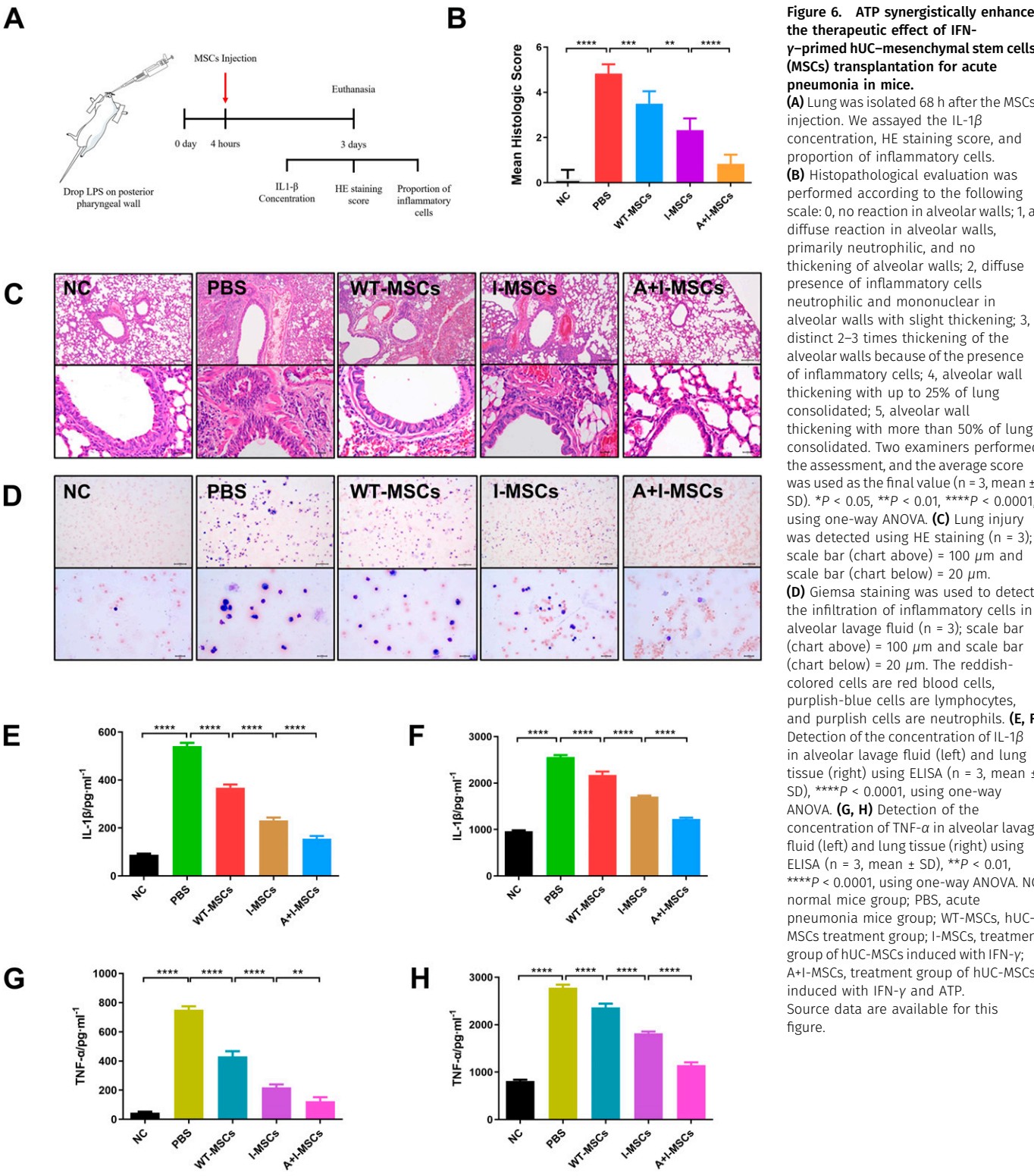

**Figure 6. ATP synergistically enhances the therapeutic effect of IFN-γ–primed hUC–mesenchymal stem cells (MSCs) transplantation for acute pneumonia in mice.**
**(A)** Lung was isolated 68 h after the MSCs injection. We assayed the IL-1β concentration, HE staining score, and proportion of inflammatory cells. **(B)** Histopathological evaluation was performed according to the following scale: 0, no reaction in alveolar walls; 1, a diffuse reaction in alveolar walls, primarily neutrophilic, and no thickening of alveolar walls; 2, diffuse presence of inflammatory cells neutrophilic and mononuclear in alveolar walls with slight thickening; 3, distinct 2–3 times thickening of the alveolar walls because of the presence of inflammatory cells; 4, alveolar wall thickening with up to 25% of lung consolidated; 5, alveolar wall thickening with more than 50% of lung consolidated. Two examiners performed the assessment, and the average score was used as the final value (n = 3, mean ± SD). *P < 0.05, **P < 0.01, ****P < 0.0001, using one-way ANOVA. **(C)** Lung injury was detected using HE staining (n = 3); scale bar (chart above) = 100 μm and scale bar (chart below) = 20 μm. **(D)** Giemsa staining was used to detect the infiltration of inflammatory cells in alveolar lavage fluid (n = 3); scale bar (chart above) = 100 μm and scale bar (chart below) = 20 μm. The reddish-colored cells are red blood cells, purplish-blue cells are lymphocytes, and purplish cells are neutrophils. **(E, F)** Detection of the concentration of IL-1β in alveolar lavage fluid (left) and lung tissue (right) using ELISA (n = 3, mean ± SD), ****P < 0.0001, using one-way ANOVA. **(G, H)** Detection of the concentration of TNF-α in alveolar lavage fluid (left) and lung tissue (right) using ELISA (n = 3, mean ± SD), **P < 0.01, ****P < 0.0001, using one-way ANOVA. NC, normal mice group; PBS, acute pneumonia mice group; WT-MSCs, hUC-MSCs treatment group; I-MSCs, treatment group of hUC-MSCs induced with IFN-γ; A+I-MSCs, treatment group of hUC-MSCs induced with IFN-γ and ATP.
Source data are available for this figure.

cell infiltration and reduced IL-1β and TNF-α levels (Fig 6). Globally, the COVID-19 pandemic is still rampant. Studies have reported that MSCs can significantly improve the clinical symptoms of patients' and save lives (Leng et al, 2020). Nevertheless, MSCs still frequently exhibit various negative effects in clinical applications, leading to individual differences in efficacy. Our study provides an explicit reference for reducing the adverse effects of MSCs and improving their therapeutic effects.

In summary, we have addressed the critical role of glucose metabolism in IFN-γ–licensed immunosuppressive hUC-MSCs. We provide evidence that ATP enhances its immunoregulatory function to some extent by promoting STAT phosphorylation. Furthermore, in an acute pneumonia mouse model, we demonstrated the safety and effectiveness of hUC-MSCs pretreated with a combination of ATP and IFN-γ in vivo. Briefly, our findings revealed a novel possible mechanism of cross talk between glucose metabolism, mitochondrial oxidation, and immunomodulatory functions in IFN-γ–licensed hUC-MSCs (Fig 7). However, our study has some limitations. We only explored the role of glucose metabolism in the immunomodulatory function of IFN-γ–primed hUC-MSCs but did not address the cross talk between overall energy metabolism and immunoregulation. In further research, we aim to use more specific inhibitory methods for lactic acid glycolysis, aerobic oxidation, other energy metabolism, and ATP production to address the cross talk mechanism between overall energy metabolism and immunoregulation of hUC-MSCs and conduct clinical trials to further verify the therapeutic effect of hUC-MSCs induced by ATP and IFN-γ in vivo.

Altogether, the findings of this study provide a novel theoretical basis for the comprehensive understanding of the relationship between the plasticity of MSC immunomodulation and metabolism and provide a new strategy for increasing intracellular ATP levels to improve the therapeutic efficacy of MSCs in inflammatory diseases.

# Materials and Methods

### Source and preparation of hUC-MSCs

The hUC-MSCs were provided by the FuMei Stem Cell Biotechnology Company and were isolated and cultured as previously described (He et al, 2018). All MSCs used in the experiment were derived from passages 5–10 after thorough characterization using flow cytometry.

For immune polarization experiments, the hUC-MSCs were plated in six-well plates or 75-cm$^2$ flasks at a density of 8,000 cells/cm$^2$ and incubated overnight. The following day, cells were pretreated with 6 $\mu$M antimycin A (ENZO), 2 $\mu$M rotenone, 10 mM 2-DG, 4 $\mu$M LDH-IN-1, 1 $\mu$M ruxolitinib, 200 $\mu$M CPI-613, 10 $\mu$M KPLH-1130 (MCE), or dimethyl sulfoxide (DMSO) (Solarbio) as a vehicle control (DMSO:media = 1:5,000) for 1 h. They were then either stimulated with IFN-γ (40 ng/ml) (PeproTech) or left unstimulated and incubated at 37°C and 5% $CO_2$ for 24 h until sample collection.

### Analysis of hUC-MSCs metabolism

Real-time assays of ECAR and OCR were performed using an XFe-96 Extracellular Flux Analyzer (Seahorse Bioscience). The hUC-MSCs were seeded on XFe-96 plates at a density of 2 × 10$^4$ cells/well and incubated overnight at 37°C and 5% $CO_2$ in the presence or absence of IFN-γ (40 ng/ml). ECAR was measured in XF medium under basal conditions and responded to 10 mM glucose and 2 $\mu$M oligomycin. Basal glycolysis was calculated after glucose injection (subtracting the baseline value). Maximal glycolysis was measured after oligomycin injection (subtracting the baseline value).

OCR was measured in XF media (non-buffered Dulbecco's Modified Eagle Medium, containing 10 mM glucose, 2 mM L-glutamine, and 1 mM sodium pyruvate), under basal conditions and in response to 2 $\mu$M oligomycin, 0.5 $\mu$M carbonylcyanide-4-(trifluoromethoxy)-phenylhydrazone (FCCP), and 0.5 $\mu$M of antimycin A and rotenone (all chemicals from Agilent). Basal OCR was calculated as the difference between baseline measurements and antimycin A/rotenone–induced OCR, and maximal OCR was measured as the difference between FCCP-induced OCR and antimycin A/rotenone–induced OCR. Relative maximal OCR was calculated as the difference between maximal OCR and basal OCR. Experiments with the Seahorse system were performed under the following assay conditions: 3 min mixture, 3 min wait, and 3 min measurement. Metabolic parameters were then calculated, and data were expressed as mean ± SD from three independent experiments, and statistical differences were evaluated using unpaired t tests.

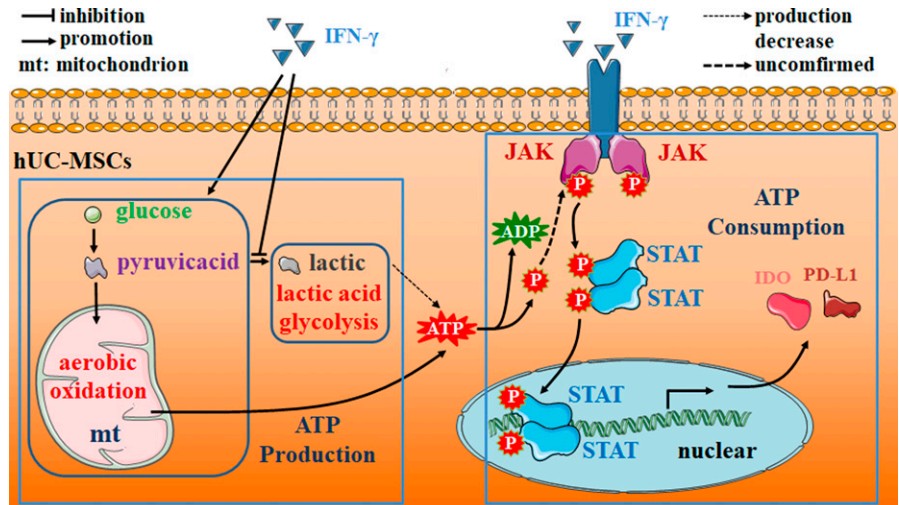

**Figure 7. Molecular mechanism of cross talk between glucose metabolism and immunosuppressive reprogramming in IFN-γ–primed hUC–mesenchymal stem cells.**

IFN-γ inhibits lactic acid glycolysis and promotes aerobic oxidation of mitochondria. ATP concentration changes continuously along with the variety of production and consumption. Furthermore, ATP can synergistically enhance the phosphorylation of JAK-STAT signaling induced by IFN-γ, thus promoting the transcription of anti-inflammatory factors IDO and PD-L1 and strengthening the immunosuppressive effect of hUC–mesenchymal stem cells.

## Metabolite detection and analysis

The hUC-MSCs were washed twice with precooled PBS. Metabolites were extracted with 1 ml cold methanol/acetonitrile (1:1, vol/vol) to remove the protein. The mixture was centrifuged for 20 min (14,000$g$, 4°C). The supernatant was dried using a vacuum centrifuge. The samples were redissolved in 100 $\mu$l acetonitrile/water (1:1, vol/vol) solvent for liquid chromatography–mass spectrometry (LC–MS) analysis. Targeted metabolomic profiling of the cells was performed using ultrahigh-performance liquid chromatography (Agilent 1290 Infinity LC) coupled with quadrupole time-of-flight mass spectrometry (UHPLC-QTOF/MS) at Shanghai Applied Protein Technology Co., Ltd. Tricarboxylic acid cycle-derived and glycolytic metabolites were detected by UHPLC using an Agilent 1290 Infinity LC column coupled to a 5500 QTRAP system (AB SCIEX) at Shanghai Applied Protein Technology Co., Ltd.

## mRNA sequencing and data analysis

For mRNA sequencing, total RNA was extracted from hUC-MSCs untreated or treated with IFN-γ (40 ng/ml) for 24 h. Total RNA was purified using QIAzol lysis reagent and the RNeasy Plus Micro kit (QIAGEN, GER). According to the manufacturer's protocol, poly(A) mRNA was isolated from total RNA fragments using the Illumina TruSeq Stranded mRNA LT Sample Prep Kit with poly T oligo-attached magnetic beads. The adapter-ligated libraries were sequenced using the Illumina HiSeq 2500 system (Illumina).

## Transfection with siRNA

The hUC-MSCs were transfected with control siRNA or siRNA against PDHA1, purchased from Ribobio (Life Technologies). The hUC-MSCs were seeded in a six-well plate (4 × 10$^4$ cells/well) and transfected with siRNA (50 nM) and 6 $\mu$l Lipofectamine RNAiMAX (Invitrogen) in 3 ml Opti-MEM (Gibco, GER). The silencing efficacy was confirmed after 24 h using real-time polymerase chain reaction (RT-PCR) and Western blotting.

## ATP treatment

The hUC-MSCs were permeabilized by the pore-forming toxin streptolysin O (SLO) (Abcam) from *Escherichia coli*, which allowed for the delivery of molecules of up to 100 kD mass to the cytosol. Briefly, SLO solution was added to Hank's balanced salt solution (Hyclone) without Ca$^{2+}$ containing 30 mM N-2-hydroxyethylpiperazine-N-ethane-sulphonic acid (Solarbio) to prepare the permeabilization solution. The SLO concentration was 100 ng/ml. The hUC-MSCs were incubated in permeabilization solution at 37°C in the presence of ATP (2 mM). After 15 min of incubation, the permeabilization solution was removed, and a fresh culture medium was added to the cells, followed by stimulation with IFN-γ for a specified time (0.5, 1.0, and 24 h). The cells were subsequently lysed for Western blotting analysis.

## Western blot

The hUC-MSCs were lysed in radioimmunoprecipitation assay buffer precooled on ice. Protein quantification of the cell lysates was performed using a bicinchoninic acid protein quantification kit (Vazyme). Equal quantities of protein were separated using 12% sodium dodecyl sulfate–polyacrylamide gel electrophoresis and transferred to a polyvinylidene fluoride membrane. The membranes were blocked with 5% nonfat milk in TBST solution (0.05% Tween 20 in Tris-buffered saline) and then incubated overnight at 4°C with primary antibodies. The next day, the blots were washed with TBST, incubated with the appropriate horseradish peroxidase–conjugated secondary antibodies (1:1,000) for 2 h at room temperature, washed again with TBST, and then developed with ECL Western blotting substrate (Millipore). Antibodies against the following proteins were purchased from CST: IDO, PD-L1, JAK2, phospho-JAK2, STAT-1, phospho-STAT-1, STAT-2, phospho-STAT-2, STAT-3, phospho-STAT-3, and β-actin.

## T-cell proliferation–suppression assay

The hUC-MSCs were seeded in 12-well plates at a density of 5 × 10$^4$ cells/well and pretreated with IFN-γ (40 ng/ml) and ATP (2 mM), as described above. hPBMCs (1 × 10$^6$ cells/ml) were activated using anti-CD3/CD28 antibodies for 48 h and then cultured with IL-2 (200 U/ml) alone for 48 h. The hPBMCs were incubated with 5 $\mu$M CFSE (Beyotime) for 20 min at a concentration of 10$^6$ cells/ml. They were protected from light during the entire procedure. After labeling, hPBMCs were stimulated with 200 IU/ml IL-2 (Acrobiosystems) in T-cell culture medium (RPMI 1640 medium supplemented with 10% FBS, 1% penicillin, and 1% glutamine) and co-cultured with or without pretreated hUC-MSCs (hUC-MSC: hPBMC ratio = 1:10) at 37°C and 5% CO$_2$. After 3 d, hPBMCs were collected, and CD3$^+$ T-cell proliferation was measured by flow cytometry as a decrease in CFSE intensity because of cell division from the original population. APC-CD3 antibody was purchased from BD Biosciences.

## Acute pneumonia mouse model

Female BALB/c mice (19–20 g) were provided standard rodent chow and water ad libitum. Mice were housed for a minimum of 2 wk in a quarantine room with a 12:12 h light–dark cycle before being used in the experiments.

The mice were anesthetized with chloral hydrate (100 g/l, i.v.). Control mice received 60 $\mu$l PBS (pH 7.2). During anesthesia, forceps were used to pull the mouse's tongue, 50 $\mu$l LPS (1 g/l) was added to the back wall of the pharynx, and the mouse's nose was immediately pinched for 20 s; then the tongue and nose were loosened, and the mouse was placed in the cage. It woke up naturally. In this study, a mouse model of acute pneumonia was developed. After treatment with LPS for 4 h, the mice received a caudal vein injection of 1 × 10$^6$ hUC-MSCs or PBS. After treatment with LPS for 3 d, the mice were euthanized.

Procedures involving mice and their care were performed according to the approved protocols and animal welfare regulations of the Animal Care and Use Committee at the Army Medical University and EU (Directive 2010/63/EU) ethical guidelines. They complied with the guidelines of the National Institutes of Health Guide for the Care and Use of Laboratory Animals.

## Analysis of histopathology

The lungs were then fixed in 4% paraformaldehyde. After fixation, a 2-mm thick midsagittal section of left and right lung lobes was

prepared and placed into a tissue-processing cassette. Sections in cassettes infiltrated with paraffin were embedded using a Tissue Tek embedding center, sectioned at 4 mm using a Reichert microtome, and stained with HE using standard methods. A board-certified veterinary pathologist conducted a microscopic examination of the stained lung sections. Scores were assigned to each lung section based on the degree of endothelial damage, severity of inflammation, percentage of neutrophils in the reaction, number of inflammatory cells in the alveoli, and number of neutrophils in the bronchioles.

### Concentration assay of ATP and IL-1$\beta$

The concentrations of ATP and IL-1$\beta$ were determined using ATP assay kits (Beyotime) and human IL-1$\beta$ ELISA kits (Boster). Experiments were performed per the manufacturer's instructions.

### Transfection with siRNA, Western blot, and T-cell proliferation–suppression assay

Refer to the Materials and Methods section of the main manuscript section for detailed methodology.

### Extracellular glucose and lactate assay

The concentration of extracellular glucose and lactate was determined using glucose test kits (Beyotime) and lactate test kits (Solarbio). Experiments were performed as per the manufacturer's instructions.

### Real-time PCR

Total RNA was extracted from the cell lysates using a high-purity total RNA extraction kit (Bio Tek) according to the manufacturers' instructions. RNA was reverse-transcribed into cDNA using the Hiscript II Q RT SuperMix (Vazyme) according to the manufacturer's instructions. Relative mRNA levels were quantified by RT-PCR using the ChamQ Universal SYBR qPCR Master Mix (Vazyme) on a C1000 Touch Thermal Cycler (Bio-Rad). Target gene expression was normalized to that of $\beta$-actin. The primer sequences used were as follows:

PDHA1 (5′-GCACCTGAAGGAGACTTGGG-3′/5′-CTCTGCTTGCCGGCTTCT-3′)
$\beta$-Actin (5′-CATTCCAAATATGAGATGCGTTGT-3′/5′-TGTGGACTTGGGA-GAGGACT-3′)

### Flow cytometry

The hUC-MSCs were seeded in a six-well plate (4 × 10$^4$ cells/plate). After treatment with ATP and IFN-$\gamma$ for 24 h, the cells were collected to be stained with 7AAD and annexin V according to the manufacturer's instructions (BD Bioscience) and analyzed using flow cytometry.

### CCK-8 cell counting assay

The cell viability was determined using CCK-8 cell counting kits (Vazyme). Experiments were performed as per the manufacturer's instructions.

### Statistical analysis

Unless otherwise noted, all experiments were repeated at least three times, and representative data were reported. Data are expressed as the mean ± SD of the samples. Statistical analysis (ANOVA and $t$ test) was performed using GraphPad Prism 6 software; $P < 0.05$ was considered to indicate statistical significance. NS: $P > 0.05$; *$P < 0.05$; **$P < 0.01$; ***$P < 0.001$; ****$P < 0.0001$.

# Supplementary Information

# Acknowledgements

The authors would like to thank all involved staff and students for their commitment and effort, without whom the study could not have been successfully completed. Funding: This study was supported by the National Natural Science Foundation of China (Nos. 82000105 and 81871568); Military Biosafety Project (No. 19SWAQ18); and Military Research and Development Program (Nos. 2019CXJSB017 and 2019CXJSC033).

### Author Contributions

M Yao: resources, data curation, software, formal analysis, validation, and writing—original draft.
Z Chen: data curation, formal analysis, investigation, visualization, and methodology.
X He: conceptualization, formal analysis, visualization, methodology, and writing—review and editing.
J Long: resources, data curation, supervision, and funding acquisition.
X Xia: data curation, formal analysis, validation, visualization, and project administration.
Z Li: resources, formal analysis, supervision, investigation, and project administration.
Y Yang: resources, data curation, investigation, and visualization.
L Ao: data curation, formal analysis, validation, investigation, methodology, and project administration.
W Xing: resources, software, supervision, investigation, and project administration.
Q Lian: conceptualization, data curation, formal analysis, project administration, and writing—review and editing.
H Liang: conceptualization, resources, supervision, funding acquisition, investigation, methodology, project administration, and writing—review and editing.
X Xu: conceptualization, resources, software, supervision, funding acquisition, investigation, visualization, methodology, project administration, and writing—review and editing.

### Conflict of Interest Statement

The authors declare that they have no conflict of interest.

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
