## [Reviewer comments · Life Science Alliance]

Life Science Alliance

Crosstalk between Glucose Metabolism and Immunosuppression in IFN- γ -primed Mesenchymal Stem Cells

Mengwei Yao, Zhuo Chen, Xiao He, Jiaoyue Long, Xuwei Xia, Zhan Li, Yu Yang, Luoquan Ao, Wei Xing, Qizhou Lian, Huaping Liang, and Xiang Xu

DOI: <https://doi.org/10.26508/lsa.202201493>

Corresponding author(s): Xiang Xu, The Army Medical University, Chongqing

Review Timeline:

Submission Date:	2022-04-20
Editorial Decision:	2022-06-03
Revision Received:	2022-07-24
Editorial Decision:	2022-08-23
Revision Received:	2022-08-30
Accepted:	2022-08-30

Scientific Editor: Novella Guidi

Transaction Report:

June 3, 2022

Re: Life Science Alliance manuscript #LSA-2022-01493-T

Prof. Xiang Xu
University of Pittsburgh Cancer Institute
Pharmacology and Chemical Biology
5112 Centre Avenue
Suite 2.35
Pittsburgh, PA 15143

Dear Dr. Xu,

Thank you for submitting your manuscript entitled "Crosstalk between Glucose Metabolism and Immunosuppression in Mesenchymal Stem Cell" to Life Science Alliance. The manuscript was assessed by expert reviewers, whose comments are appended to this letter. We invite you to submit a revised manuscript addressing the Reviewer comments.

Thank you for this interesting contribution to Life Science Alliance. We are looking forward to receiving your revised manuscript.

Sincerely,

B. MANUSCRIPT ORGANIZATION AND FORMATTING:

Reviewer #1 (Comments to the Authors (Required)):

1. A short summary of the paper, including description of the advance offered to the field.

Yao et al, has aimed to investigate in IFN- γ and glucose metabolism in umbilical cord mesenchymal stem cells (hUC-MSCs) and they showed that glucose metabolism regulates the immunosuppressive function of human umbilical cord MSCs 31 (hUC-MSCs) induced by IFN- γ through the JAK-STAT pathway. This work is so important to investigate IFN- γ to find immunosuppressive function and glucose metabolism and provide a novel targeting strategy for MSCs. However, some parts should be revised by authors.

2. For each main point of the paper, please indicate if the data are strongly supportive. If not, explicitly state the additional experiments essential to support the claims made and the timeframe that these would require.

Manuscript looks fine in general however some sentences need to be revised. Overall comment, study looks interesting, but discussion part should be revised, and figures and data should be explained more by discussing with other relevant publications.

3. Lastly, indicate any additional issues you feel should be addressed (text changes, data presentation, statistics etc.).

I would like to mention that all methods which have been used in the manuscript is very well to investigate the question.

Not all figures were referred in the text in the same way. Please make sure all figures were discussed and referred in the text and referred as Fig or Figure.

Introduction part needs to be more straight forward written to transmit the idea in a better view.

Aim of the study is not clearly stated

Page 3 Line 56, sentence should be revised.

Abstract should be explained and written detailed way

Please indicate what is lack of this study.

The English language of the article should be revised.

Discussion part needs to be rearranged in order.

Authors may think of using different word then "Therefore"

What would be done next to confirm and extend the data?

Reviewer #2 (Comments to the Authors (Required)):

The revised paper shows strong data on how human multipotent umbilical cord cells (hu-MSC) stimulated with IFN gamma and ATP signify a potential solution for the immune responsiveness provoked when hu-MSC are implanted in the target areas of a model of inflammatory disorder. To this effect, the paper also highlights a potential cellular mechanism based on the modulation of the lactic and aerobic glycolysis in hu-MSC by the stimulators which is relevant as it shows to differ from other types of multipotent stem cells. In general terms, the manuscript is focused on the effects of stimulation of IFN gamma and, IFN gamma + ATP compared to non-stimulated controls. I'd suggest the authors to use an analogue or antagonist to the IFN treatment, if available, to make a greater emphasis on the results. This as one can speculate the mechanical stimulation may cause some of the effects observed. If possible, to ask for 1 experiment with a compound of either kind? In addition, metabolic measurements after rapid change from hyperglycaemic culture media to specific media for Seahorse analysis may alter the cellular response. Is it possible for the authors to estimate the changes in glycolytic capacity by measurements of extracellular glucose and lactate from the culture media of stimulated and control conditions E.g., using glucose and lactate kits?

I'd put forward this manuscript for consideration of publication after the authors address at the minimum the following points:

Show data on cellular survival after treatment with IFN gamma and IFN gamma + ATP

Lines 132-134:

Figure 1 C & D: The authors suggest that the treatment inhibit the percentage to T cell proliferation when cocultured yet, the plots are based on MFI? I'd suggest this to be revised for more accuracy and perhaps either correcting the labels (D) to MFI or

to plot using cell counts

Lines 146-149:

Figure 2 B & D: The authors examined the expression of all the isoforms of PDK and ENO, are all relevant to the same extent in the cell type of interest? One can speculate that the expression of some isoforms does not correlate with their kinetics of activity in a particular cell type. Perhaps this should be revised to avoid overinterpretation of data that is not relevant. Please the graphs axis needs corrections from ratio to ratio.

Lines 151 to 154:

Figures 2 E to H: The authors showed that the IFN treatment increases the baseline of OCR and ECAR. Although this is significant compared to the controls, I'd suggest the authors show the ratio of changes in the maximal OCR instead of the neat maximal OCR.

Lines 166 to 171:

Figure 3: The authors showed that inhibiting mitochondrial activity significantly reduces the expression of anti-inflammatory molecules. Although this is a true effect, the authors are not examining mitochondrial beta-oxidation thus, I'd suggest being caution in the conclusions from these results. E.g., cytosolic glycolysis is not relevant whilst mitochondrial respiration either from glucose or fatty acid oxidation is essential for the anti-inflammatory function.

Lines 210-211:

Figure 5 H to J: Similar comment than for figure 1. If the percentage is estimated based on the MFI, I'd change it to actual cell count for more accuracy. Otherwise, I'll interpret it as MFI meaning, the larger accumulation of a marker not always suggests more cells but higher intensity of absorption of that dye which can come by e.g., larger cell size.

The largest result was obtained after challenge of hu-MSC with IFN + ATP. I'd put this in the summary figure to avoid misleading as IFN alone treatment did not promote an increase in the ATP levels nor a significantly clear change in glycolysis.

Lines 283-284:

Ideas need paraphrasing for better clarity.

Line 299:

The statement is too strong for what has been proved in this research. I'd change this to a less conclusive idea such as the one expressed in the paragraph before.

Line 302:

I suggest a change in the sentence to e.g., between glucose metabolism, mitochondrial oxidation, and immunomodulatory functions...

Line 305:

Is targeting glucose metabolism the best approach or perhaps increasing intracellular levels of ATP?

Please find the following responses to the referee comments:

● **Response to Reviewer #1:**

Comments 1: A short summary of the paper, including description of the advance offered to the field.

Yao et al, has aimed to investigate in IFN- γ and glucose metabolism in umbilical cord mesenchymal stem cells (hUC-MSCs) and they showed that glucose metabolism regulates the immunosuppressive function of human umbilical cord MSCs 31 (hUC-MSCs) induced by IFN- γ through the JAK-STAT pathway. This work is so important to investigate IFN- γ to find immunosuppressive function and glucose metabolism and provide a novel targeting strategy for MSCs. However, some parts should be revised by authors.

Response: We would greatly appreciate you for your valuable comments and positive evaluation. According to your valuable comments, we have revised our manuscripts by point to point.

Comments 2: For each main point of the paper, please indicate if the data are strongly supportive. If not, explicitly state the additional experiments essential to support the claims made and the timeframe that these would require.

Response: We would highly appreciate you for this critical comment to improve our manuscript. Our data shown in text basically supports our conclusion, and we revised some conclusion in new manuscript. Absolutely, there are still some deficiencies in our research. According to your valuable comments, we added the survival detection of hUC-MSCs treated by combination of ATP and IFN- γ , and the results shown in Fig S4. In addition, we also added the measurement experiment about extracellular glucose and lactate in the metabolic measurements for seahorse analysis, and the results were showed in in Fig S2.

Comments 3: Manuscript looks fine in general however some sentences need to be

revised. Overall comment, study looks interesting, but discussion part should be revised, and figures and data should be explained more by discussing with other relevant publications.

Response: We would greatly appreciate you for your positive evaluation and valuable comments. In revised manuscript, many sentences have been revised. Discussion section has been revised. The results have been explained through citing more other relevant publications.

Comments 4: Lastly, indicate any additional issues you feel should be addressed (text changes, data presentation, statistics etc.).

I would like to mention that all methods which have been used in the manuscript is very well to investigate the question. Not all figures were referred in the text in the same way. Please make sure all figures were discussed and referred in the text and referred as Fig or Figure.

Response: we would greatly appreciate you for your valuable comments. According to your comments, we changed the text, data and statistics presentation in the same way, and all figures have been discussed and referred in the text in the same way.

Comments 5: Introduction part needs to be more straight forward written to transmit the idea in a better view.

Response: According to your valuable comments, we revised the introduction section.

Comments 6: Aim of the study is not clearly stated.

Response: According to your valuable comments, we revised the description of aim of the study in the abstract and introduction section.

Comments 7: Page 3 Line 56, sentence should be revised.

Response: We would greatly appreciate the reviewer for your careful and valuable

comments, this sentence has been revised from “However, the current clinical efficacy of MSC transplantation (MSCT) still displays substantial individual differences, even in the same autoimmune disease.” to “However, the current clinical efficacy of hMSCs is still variable, even among individuals with the same autoimmune disease.”

Comments 8: Abstract should be explained and written detailed way.

Response: We would greatly appreciate the reviewer for your careful and valuable comments. We have corrected the abstract in detailed way according to your comments.

Comments 9: Please indicate what is lack of this study.

Response: We would greatly appreciate you for your valuable comments. we added the lack of this study in the discussion section. “However, our study has some limitations. We only explored the role of glucose metabolism in the immunomodulatory function of IFN- γ -primed hUC-MSCs but did not address the crosstalk between overall energy metabolism and immunoregulation.”

Comments 10: The English language of the article should be revised. Discussion part needs to be rearranged in order. Authors may think of using different word then "Therefore". What would be done next to confirm and extend the data?

Response: We would highly appreciate you for your valuable comments. The English language and discussion of the article have been revised. In addition, for confirming and extending the data, we will use more specific methods to inhibit lactic acid glycolysis, aerobic oxidation and ATP production. We will also conduct clinical trials to further verify the therapeutic effect of hUC-MSCs induced by ATP and IFN- γ *in vivo*. These contents have added in the discussion section of revised manuscript.

● **Response to Reviewer #2:**

Comments 1: The revised paper shows strong data on how human multipotent umbilical cord cells (hu-MSC) stimulated with IFN gamma and ATP signify a potential solution for the immune responsiveness provoked when hu-MSC are implanted in the target areas of a model of inflammatory disorder. To this effect, the paper also highlights a potential cellular mechanism based on the modulation of the lactic and aerobic glycolysis in hu-MSC by the stimulators which is relevant as it shows to differ from other types of multipotent stem cells. In general terms, the manuscript is focused on the effects of stimulation of IFN gamma and, IFN gamma + ATP compared to non-stimulated controls. I'd suggest the authors to use an analogue or antagonist to the IFN treatment, if available, to make a greater emphasis on the results. This as one can speculate the mechanical stimulation may cause some of the effects observed. If possible, to ask for 1 experiment with a compound of either kind? In addition, metabolic measurements after rapid change from hyperglycaemic culture media to specific media for seahorse analysis may alter the cellular response. Is it possible for the authors to estimate the changes in glycolytic capacity by measurements of extracellular glucose and lactate from the culture media of stimulated and control conditions E.g., using glucose and lactate kits?

Response: We would greatly appreciate you for your valuable and positive comments. Exactly, if we can use an analogue or antagonist to the IFN treatment, the data will be more perfect. According to your valuable comments, we seek information about analogue or antagonist of IFN- γ . However, Unfortunately, we not find the information about analogue or antagonist of IFN- γ . In addition, as we known, IFN- γ -induced the anti-inflammatory function of hUC-MSCs had been confirmed many research groups. In our previously study, we demonstrated that the anti-inflammatory function of IFN-gammaR(-/-) MSCs show significantly decrease (He,X,et al. Ann Rheum Dis. 2020 Oct; 79(10): 1298-1304.). Therefore, we did not add experiments about analogue or antagonist of IFN- γ . In our next research, we will do it if we get analogue or

antagonist of IFN- γ .

About metabolic measurements, indeed, we changed the culture medium when using Seahorse analysis. According to your valuable comments, we added experiments to measure the changes of extracellular glucose and lactate from the culture media of stimulated and control conditions shown in Fig S2.

Comments 2: I'd put forward this manuscript for consideration of publication after the authors address at the minimum the following points:

Show data on cellular survival after treatment with IFN γ and IFN γ + ATP.

Response: According to your valuable comments, we added the cellular survival data shown in Fig S4.

Comments 3: Lines 132-134: Figure 1 C & D: The authors suggest that the treatment inhibits the percentage of T cell proliferation when cocultured yet, the plots are based on MFI? I'd suggest this to be revised for more accuracy and perhaps either correcting the labels (D) to MFI or to plot using cell counts.

Response: We would highly appreciate the reviewer for your careful and valuable comments. According to your valuable advice, we revised Fig 1C and D using MFI and cell counts data shown in Fig S1.

Comments 4: Lines 146-149: Figure 2 B & D: The authors examined the expression of all the isoforms of PDK and ENO, are all relevant to the same extent in the cell type of interest? One can speculate that the expression of some isoforms does not correlate with their kinetics of activity in a particular cell type. Perhaps this should be revised to avoid overinterpretation of data that is not relevant. Please the graphs axis needs corrections from ratio to ratio.

Response: We would highly appreciate the reviewer for your careful and valuable comments. At present, there is no study to confirm which isoforms of glucose

metabolism related enzymes are regulated by IFN- γ in hUC-MSCs. Therefore, we detected the isoforms expression of all related enzymes in order to confirm whether IFN- γ affects the process of glucose metabolism by regulating the expression of enzymes, and identify which isoforms are the key metabolic enzymes affected by IFN- γ . Ratio in Fig 2B and D has been corrected to ratio.

Comments 5: Lines 151 to 154: Figures 2 E to H: The authors showed that the IFN treatment increases the baseline of OCR and ECAR. Although this is significant compared to the controls, I'd suggest the authors show the ratio of changes in the maximal OCR instead of the neat maximal OCR.

Response: We would highly appreciate the reviewer for your valuable comments. Exactly, the change of the maximum OCR relative to the basic value can really reflect the change of glucose metabolism. According to your valuable comments, we added a plot in Fig 2.

Comments 6: Lines 166 to 171: Figure 3: The authors showed that inhibiting mitochondrial activity significantly reduces the expression of anti-inflammatory molecules. Although this is a true effect, the authors are not examining mitochondrial beta-oxidation thus, I'd suggest being caution in the conclusions from these results. E.g., cytosolic glycolysis is not relevant whilst mitochondrial respiration either from glucose or fatty acid oxidation is essential for the anti-inflammatory function.

Response: We would highly appreciate the reviewer for your valuable comments. Exactly, you are right. We correct the conclusion according to your comments.

Comments 7: Lines 210-211: Figure 5 H to J: Similar comment than for figure 1. If the percentage is estimated based on the MFI, I'd change it to actual cell count for more accuracy. Otherwise, I'll interpret it as MFI meaning, the larger accumulation of a marker not always suggests more cells but higher intensity of absorption of that dye which can come by e.g., larger cell size.

Response: We would highly appreciate the reviewer for your careful and valuable comments. According to your valuable advice, we revised Fig 5 H to J using MFI and cell counts data shown in Fig S5.

Comments 8: The largest result was obtained after challenge of hu-MSC with IFN + ATP. I'd put this in the summary figure to avoid misleading as IFN alone treatment did not promote an increase in the ATP levels nor a significantly clear change in glycolysis.

Response: The summary figure has been revised for avoiding misleading. Indeed, in our study, ATP concentration changes continuously along with the variety of production and consumption. So IFN- γ alone treatment did not promote an increase in the ATP levels.

Comments 9: Lines 283-284: Ideas need paraphrasing for better clarity.

Response: It has been revised from “A previous study has shown that there is an ATP-binding domain in JAK2. Mutation of the amino acid at the binding site affects phosphorylation of the JAK-STAT pathway (Hammarén et al, 2015).” to “A previous study has shown that ATP can bind to a pseudokinase domain in JAK2 and participate in the hyperactivation of JAK2. Mutation of the amino acid at the binding site ameliorates hyperphosphorylation of JAK2 in mutants V617F (Hammarén et al, 2015).”

Comments 10: Line 299: The statement is too strong for what has been proved in this research. I'd change this to a less conclusive idea such as the one expressed in the paragraph before.

Response: According to your valuable comments, the statement has been revised from “We confirmed that ATP enhances its immunoregulatory function by promoting STAT phosphorylation in molecular mechanisms.” to “We provide evidence that ATP enhances its immunoregulatory function to some extent by promoting STAT

phosphorylation.”

Comments 11: Line 302: I suggest a change in the sentence to e.g., between glucose metabolism, mitochondrial oxidation, and immunomodulatory functions...

Response: We would greatly appreciate the reviewer for your careful and valuable comments. The sentence has been revised from “Briefly, our findings revealed a novel possible mechanism of crosstalk between glucose metabolism and immunomodulatory functions in IFN- γ -licensed hUC-MSCs.” to “Briefly, our findings revealed a novel possible mechanism of crosstalk between glucose metabolism, mitochondrial oxidation and immunomodulatory functions in IFN- γ -licensed hUC-MSCs.”

Comments 12: Line 305: Is targeting glucose metabolism the best approach or perhaps increasing intracellular levels of ATP?

Response: We would highly appreciate the reviewer for your careful and valuable comments. You are right, the best and direct approach is increasing intracellular levels of ATP. According to your valuable comments, we have been revised in line 311.

We would highly appreciate you for your valuable comments once again!

August 23, 2022

RE: Life Science Alliance Manuscript #LSA-2022-01493-TR

Prof. Xiang Xu
Army Medical University
Daping Hospital
Changjiang Road
Chongqing 400042
China

Dear Dr. Xu,

Thank you for submitting your revised manuscript entitled "Crosstalk between Glucose Metabolism and Immunosuppression in IFN- γ -primed Mesenchymal Stem Cells". We would be happy to publish your paper in Life Science Alliance pending final revisions necessary to meet our formatting guidelines.

- please add ORCID ID for corresponding (and secondary corresponding) author--you should have received instructions on how to do so
- please add a Summary Blurb/Alternate Abstract in our system
- the Supplemental Experiments should be incorporated into the main Materials and Methods section. there is no size limit to this section
- please list the main and supplemental figure legends at the end of the manuscript file, without the figures
- please add a callouts for all Figure 4 panels to your main manuscript text (there is a callout only for Fig 4, not A, B, C, D); please update your callouts for the Supplementary Figures in the manuscript Fig S1 (A,B), Fig S2 (A, B), Fig S3 (A, B, C); Fig S4 (A, B, C); Fig S5 (A,B))
- please add sizes next to all blots

A. FINAL FILES:

B. MANUSCRIPT ORGANIZATION AND FORMATTING:

Sincerely,

Reviewer #1 (Comments to the Authors (Required)):

Dear Authors,

Thank you for your efforts. After revisions all have been performed, manuscript looks better. I would like congratulate you for the great work you have done.

All the best,

Please find the following responses to the referee comments:

● **Response to Editor:**

Comments 1: Please add ORCID ID for corresponding (and secondary corresponding) author.

Response: We have added ORCID ID for two corresponding authors.

Comments 2: Please add a Summary Blurb/Alternate Abstract in our system.

Response: Summary Blurb/Alternate Abstract have been added in the system.

Comments 3: The Supplemental Experiments should be incorporated into the main Materials and Methods section.

Response: We have incorporated the Supplemental Experiments into the main Materials and Methods section.

Comments 4: Please list the main and supplemental figure legends at the end of the manuscript file, without the figures.

Response: We revised the main and supplemental figure legends according to your valuable evaluation.

Comments 5: Please add a callouts for all Figure 4 panels to your main manuscript text (there is a callout only for Fig 4, not A, B, C, D)

Response: According to your valuable comments, we revised the callouts of Figure 4.

Comments 6: Please update your callouts for the Supplementary Figures in the manuscript Fig S1 (A,B), Fig S2 (A, B), Fig S3 (A, B, C); Fig S4 (A, B, C); Fig S5 (A,B)

Response: According to your valuable comments, we updated callouts for the Supplementary Figures in the manuscript.

Comments 7: Please add sizes next to all blots.

Response: According to your valuable comments, we added size next to all blots.

Comments 8: We encourage our authors to provide original source data, particularly uncropped/-processed electrophoretic blots and spreadsheets for the main figures of the manuscript. If you would like to add source data, we would welcome one PDF/Excel-file per figure for this information. These files will be linked online as supplementary "Source Data" files.

Response: We would greatly appreciate you for your careful and valuable comments. We have added original source data for the main figures of the manuscript.

We would highly appreciate you for your valuable comments once again!

August 30, 2022

RE: Life Science Alliance Manuscript #LSA-2022-01493-TRR

Prof. Xiang Xu
Army Medical University
Daping Hospital
Changjiang Road
Chongqing 400042
China

Dear Dr. Xu,

Thank you for submitting your Research Article entitled "Crosstalk between Glucose Metabolism and Immunosuppression in IFN- γ -primed Mesenchymal Stem Cells". It is a pleasure to let you know that your manuscript is now accepted for publication in Life Science Alliance. Congratulations on this interesting work.

DISTRIBUTION OF MATERIALS:

Again, congratulations on a very nice paper. I hope you found the review process to be constructive and are pleased with how the manuscript was handled editorially. We look forward to future exciting submissions from your lab.

Sincerely,
